# Tailoring neuromuscular dynamics: A modeling framework for realistic sEMG simulation

**Alvaro Costa-Garcia**[1]*, **Shingo Shimoda**[2], **Akihiko Murai**[1]

**1** Research Institute on Human and Societal Augmentation, National Institute of Advanced Industrial Science and Technology (AIST), Kashiwa, Chiba, Japan, **2** Graduate School of Medicine, Nagoya University, Nagoya, Aichi, Japan

* alvaro.costagarcia@aist.go.jp

**Data availability statement:** All relevant data are within the paper and its Supporting information files. All supplementary materials

## Abstract

This study introduces an advanced computational model for simulating surface electromyography (sEMG) signals during muscle contractions. The model integrates five elements that simulate the chain of processes from motor intention to voltage variations over the skin. These elements include the motor control system, motor neurons, muscle fibers, biological tissues, and electrodes. sEMG signals were simulated for isotonic and isometric contractions under two force conditions and compared with real data obtained from elbow flexion experiments. The results demonstrate a high level of similarity between simulated and real signals, encompassing both temporal and spectral features. Additionally, the study reveals a correlation between muscle fiber type distribution and changes in the spectral distribution of the simulated signals. Potential applications of this research include the development of comprehensive sEMG databases and elucidating the relationship between sEMG signal characteristics and internal neuromuscular parameters. Future research aims to further explore these applications and enhance the model's performance by leveraging emerging technologies such as machine learning. This approach establishes a framework for simulating sEMG signals under tailored neuromuscular conditions and holds promise for advancing our understanding of muscular physiology and human motor control mechanisms.

## Introduction

Surface electromyography (sEMG) signals are widely used to infer muscle activity and movement. Physiological models have been developed to simulate sEMG signals, with numerous proposals put forth to explain the underlying neural and muscular processes that generate these signals. Early proposals include Hill's muscle model [1], Huxley and Niedergerke's sliding filament model [2], and the motor unit action potential (MUAP) model [3]. In these initial approaches, mathematical models were used to simulate the electrical and mechanical properties of simple autonomous modules such as single muscle fibers and motor units.

are available at
https://doi.org/10.57765/2000617.

**Funding:** This work was supported by the National Institute of Advanced Industrial Science and Technology (AIST), the Moonshot R&D Program under Grant JPMJMS2239 and the Japan Society for the Promotion of Science through the Kakenhi Kiban-C under Grant 25K15294. There was no additional external funding received for this study. The funders had no role in study design, data collection and analysis, decision to publish, or preparation of the manuscript.

**Competing interests:** The authors have declared that no competing interests exist.

Recently, more realistic simulations have been developed by combining previously existing models and including complex anatomical and physiological properties that affect sEMG signals [4]. For example, Peterson et. al. developed an integrative solution that combines several physiological factors to develop a motor unit pool organization, sEMG, and force generation model for sustained isometric contractions [5]. Other models, such as the one proposed by Pereira et al [6], incorporate imaging techniques to model the anatomical distribution of muscle fibers and the volume conduction properties of the biological tissues between muscle fibers and electrodes, resulting in more realistic simulation of recorded sEMG signals.

Most studies that simulate sEMG use the motor unit as the basic unit of contraction. However, this approach has limitations since it assumes that all fibers of a motor unit innervate when it is activated, without taking into account that some fibers may remain inactive depending on the fire rate. For this reason, many studies focus on simulating sustained isometric contractions, where it is reasonable to assume that all fibers of a motor unit are innervated [7–9]. However, daily activities typically involve short-term, low-intensity isometric and isotonic contractions, where motor unit firing rate is low and only slow-twitching fibers with lower activation thresholds are recruited [10].

A solution to this problem is to take the muscle fiber as the basic unit for muscle activity simulation. Each fiber is independent and only innervates once its activation threshold is reached. The goal of this work is to develop a flexible framework for simulating muscle activity that centers on the muscle fiber as a minimal module of activation. Our approach considers the muscle fiber as an independent programmable object with all its properties defined, allowing for a wide range of possible configurations. Parameters such as the fiber radius, length, intracellular conductivity, conduction velocity, spatial distribution, force generation properties, etc., can be adjusted according to the specific requirements of the task or muscle being simulated.

Moreover, the proposed muscle fibers can be flexibly grouped into motor units and muscles, and higher-level control strategies, such as the recruitment of motor units and fire rate, can also be customized. It is important to note that the main goal of this paper is not to simulate a specific muscle or activation but to provide a basic building block for simulating realistic sEMG signals.

The complexity of the motor control system and current technical limitations make it impossible to simultaneously measure both sEMG signals and all the neuromuscular parameters that produce them. This has led to various discussions within the scientific community regarding the feasibility of inferring neuromuscular parameters from the evaluation of sEMG signals [11–13]. The model presented in this paper enables the creation of a programmatic environment in which both neuromuscular parameters and the generated sEMG signal are known, providing a reasonable framework for testing the relationships between them.

The following section is organized into three parts. First, a comprehensive explanation of the model conceptualization. Second, the detailed description of each modeled element. Third, a validation stage were all the aforementioned components are integrated into a simulation of the human biceps, using specific parameter values and comparing simulated and real sEMG signals.

## Materials and methods

### General description of the model

Fig 1 shows a graphical overview of the general functioning of our model, which is based in the following five interconnected components.

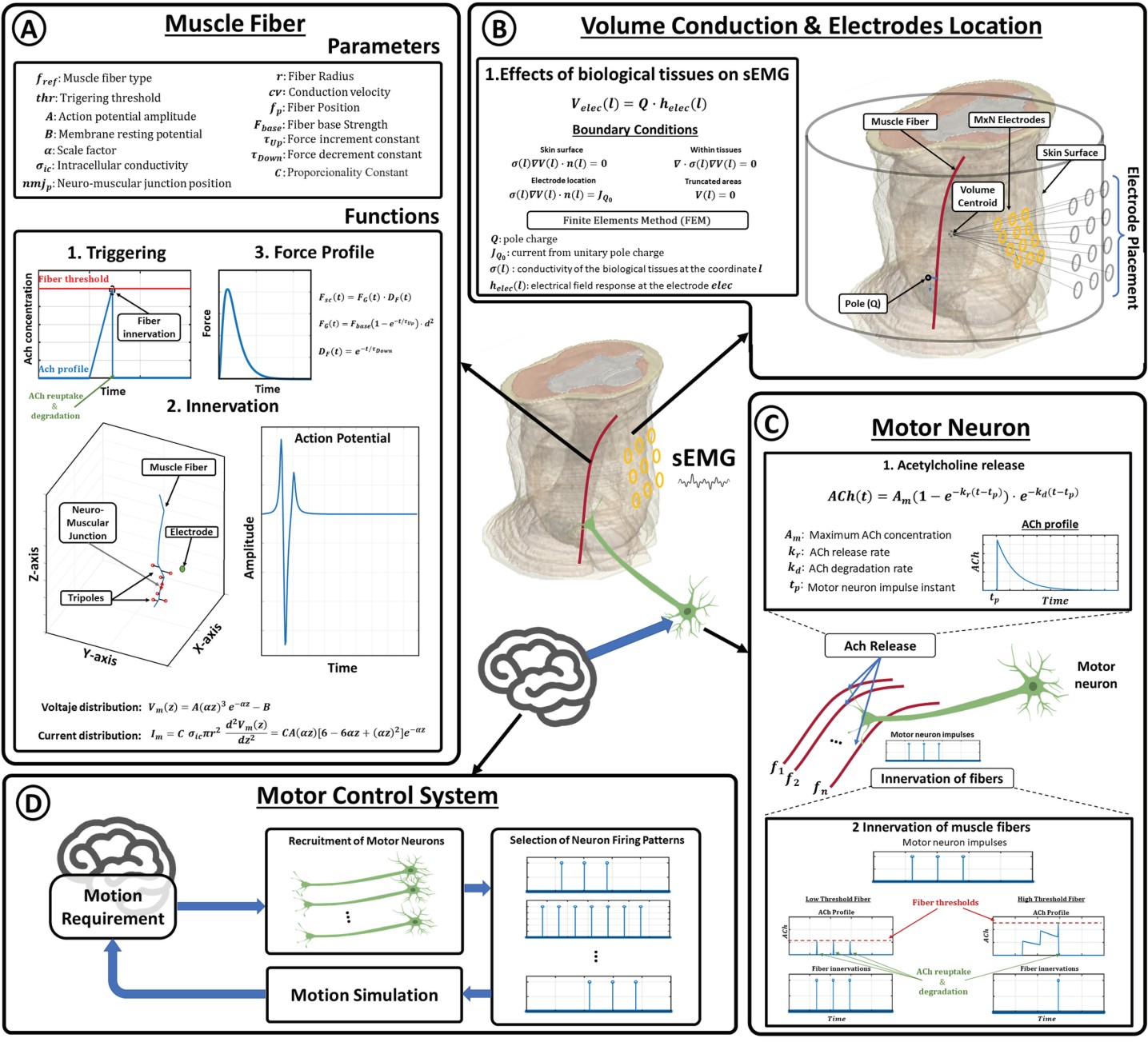

**Fig 1. Model components.** Overview of the five elements composing the proposed model: motor control system, motor neuron, muscle fiber, biological tissues volumes and electrodes.

- **Electrodes:** Defined as points in the 3D space, each one establishes a location where changes in the electromagnetic field produced by traveling membrane potentials will be measured (Fig 1B).
- **Volume Conduction Environment:** A volumetric model of the biological tissues (Fig 1B), including muscle, cancellous and cortical bone, and fat/skin, materialized as a 3D grid where each point is related to the dominant tissue in the referenced voxel. This

mesh will serve two purposes: (a) delimiting the 3D contour of the measurement area to position the electrodes realistically in 3D space, and (b) serving as an input parameter for each innervated muscle fiber to calculate modifications in the potential difference between the membrane and electrode according to the distribution of media between them.

- **Motor Neuron:** The third component is a motor neuron (Fig 1C), which receives impulses from the motor control system and converts them into profiles of acetylcholine emission, which, in turn, serves as triggering neurotransmisor for the inervation of muscle fibers based on their activation threshold.
- **Motor Control System:** Responsible for selecting which motor neurons and fire rates to apply to generate the desired movement (Fig 1D).
- **Muscle Fiber:** A programmatic object containing variables to represent a large number of physical and electrical properties of the motor fiber, as well as a set of methods to simulate the propagation of potentials through the fiber membrane and its force profile upon innervation (Fig 1A).

In the following section, each of the described components will be detailed, and in the validation section, physiological parameters will be fixed to simulate sEMG signals produced by the contraction of the biceps brachii. However, it is important to note that the objective of this model is to allow great flexibility in generating muscle activations, so the presented methods are not meant to be imposed. Each section will clarify which parts are methods of generation that could be substituted depending on the simulation objectives.

## Muscle fiber

A MuscleFiber class was developed in Matlab in order to define the properties and functions of muscle fiber object. The class includes a number of variables representing different parameters of muscle fibers. The number of variables can be modified and adapted to the needs of the simulation. For example, for the current work it was included a property indicating the type of muscle fiber ($f_{type}$). This variable was used only as a reference to define other fiber properties based on its type. The rest of parameters defined (highlighted with bold letters in the following subsections) will be used for the definition of the following muscle fiber functions.

**Innervation triggering.** A simple function triggering fiber innervation based on an acetylcholine threshold (**thr**). This function continuously evaluates external acetylcholine profiles (generated by the motor neuron) and once the acetylcholine level reaches the internal threshold of the fiber, innervation function is activated (Fig 1A1). Moreover, this function also simulates acetylcholine reuptake and degradation processes [14–16] by reducing external acetylcholine profiles to zero after innervation. In this way , for the fiber to innervate again will require new acetylcholine releases from the motor neuron. This work uses only the acetylcholine threshold as triggering mechanism, however trigger conditions can be expanded by including additional methods that simulate other neurotransmitters involved in fiber innervation [17–19].

**Potential propagation.** The simulation of potential propagation in a fiber membrane is carried out once the fiber is innervated. Its implementation is based on the well-known tripole model originally proposed by Merletti et al [20]. The voltage distribution in the fiber membrane can be defined following Rosenfalck [21] approach as:

$$V_m(z) = A(\alpha z)^3 e^{-\alpha z} - B \tag{1}$$

Here, $A$ represents the amplitude of the action potential, $B$ is the resting membrane potential, z denotes the distance along the fiber (with z = 0 at the neuro-muscular junction), and $\alpha$ is a scale factor measured in $mm^{-1}$ that can be tuned to adjust the spectral components of the simulated sEMG signals to the real sEMG bandwidth. The current distribution $I_m$ is given by the second derivative of $V_m(z)$:

$$I_m = C\sigma_{ic}\pi r^2 \frac{d^2 V_m(z)}{dz^2} \tag{2}$$

Here, $C$ is a proportionality constant to adjust the amplitud of simulated sEMG signals to real data values. Moreover, the intensity amplitude was related to the fiber intracellular conductivity ($\sigma_{ic}$) and radius ($r$) [2, 3]. The triphasic current $I_m$ can be represented as three equivalent point sources located at the centroid of each current phase along the membrane axis. As described in Merletti et al [20], for each innervation, a pair of symmetric tripole sources travels from the neuro-muscular junction ($nmj_{pos}$) to the tendon locations, represented by the start and end points of the muscle fiber. The position of the tripole sources is shifted a total of $dt \cdot cv$ samples towards each tendon every $dt$, which is defined based on the desired sampling frequency $fs$ and a constant conduction velocity ($cv$) specific to each fiber (see Fig 1A2).

To allow for a wider range of simulation conditions, two main modifications were introduced to Merletti's model. Firstly, a mechanism for pole generation and extinction was implemented. Merletti [20] and other authors [22–24] impose the existence of compensatory poles at fiber ends and innervation points. This solution largely reduces the abrupt changes in the potential and allow a clear visualization of the membrane action potential. However some voltage spikes still remain. As this work aims for the simulation of longer contractions where large amounts of potentials will be overlapped, authors introduce a slightly more sophisticated pole generation and extinction method to further reduce the remaining spikes. It involves defining a region around the neuro-muscular junction where pole amplitudes are linearly modified between zero and the desired amplitude. Similarly, an inverse linear relationship is applied to extinguish the poles in the tendons. (See S1 video supplementary material for a comparison of the methods).

The second modification incorporates a spatial distribution for each muscle fiber, defined by a series of 3D points ($fib_{pos}(x,y,z)$) connected by straight segments. The poles are displaced along these segments at a constant velocity ($cv$), resulting in a three-dimensional motion according to the spatial distribution of each fiber (refer to the S2 video supplementary material for a visualization of the 3D pole displacement).

Methods for the definition of membrane currents and poles as well as poles generation, propagation and extinction are defined separately which allow their independent modification and update without affecting the rest of model elements.

**Force generation.** A function that models muscle contraction force in response to a single innervation. To model this force, current work defines the single-innervation contraction force profile as:

$$F_{SC}(t) = F_G(t) \cdot D_F(t) \tag{3}$$

$$F_G(t) = F_{base} \cdot (1 - e^{-t/\tau_{Up}}) \cdot d^2 \tag{4}$$

$$D_F(t) = e^{-t/\tau_{Down}} \tag{5}$$

where $F_G(t)$ is the profile of force growth after a single innervation. The term $(1 - e^{-t/\tau_{Up}})$ in the formula reflects the asymptotic increase of muscle fiber force over time according to $\tau_{Up}$. $F_{base}$

is the reference force that a given fiber can produce in a single innervation and $d = 2r$ is the diameter of the muscle fiber. $D_F(t)$ incorporates the decay of force as time progresses according to the value of $\tau_{Down}$. Tuning the parameters $\tau_{Up}$, $\tau_{Down}$, $F_{base}$ and $d$ allows the generation of different force activation profiles according to the bio-mechanical properties of each muscle fiber (see Fig 1A3).

By overlapping the temporal patterns of each single activation force, it is possible to compute the force profile of a muscle fiber $f$ for each time $t$ as:

$$F_f(t) = \sum_{imp=1}^{n} F_{SC}(t - t_{imp}) \tag{6}$$

where $t_{imp}$ is the instant where the innervation $imp$ occurred for $imp = 1, 2, ...n$.

In the context of simulating muscle contractions, having access to the force profiles generated by each individual muscle fiber enables the determination of when a contraction fulfills specific force requirements for a given movement. The objective of a muscle contraction is typically to achieve a desired force level or generate a specific temporal force profile. Therefore, by knowing the force contribution and activation timing of each fiber, it becomes feasible to determine when the force requirements of the movement have been met and conclude that the recruitment of additional active fibers is not necessary.

Similar to the approach described in the previous subsection, the modular structure of the developed code allows for independent modification of the equations and parameters used to simulate the force profile, in accordance with the specific requirements of the simulation.

## Volume conduction environment

The volume conduction environment is represented as a Matlab structure that includes a mesh grid (X, Y, Z) consisting of a set of 3D points within a cuboid area where all the biological tissues are defined. Additionally, the structure contains a 'data' variable of the same size with labels indicating the dominant tissue at each grid point.

The size and distribution of biological tissues can be generated based on empirical data. In this study, magnetic resonance imagery (MRI) scans were used to establish and adapt the volumes of each biological tissue to the finite element grid defined within the volume conduction environment. The grid resolution can also be adjusted to achieve the desired balance between computational demands and accuracy.

The volume conduction environment serves the two following purposes.

**Realistic electrode placement.** By differentiating the skin area as the interface between the tissues and the exterior, the volume of the biological tissues is utilized as a reference for electrode placement, ensuring that the electrodes are positioned on the skin surface.

**Effects on measured potential.** The electric potential $V_{elec}^Q(l)$ generated on the electrode *elec* by a charge $Q$ located in the 3D coordinate $l$ was calculated according to the formulation proposed by Pereira et al. [6] using the following equation:

$$V_{elec}^Q(l) = Q \cdot h_{elec}(l) \tag{7}$$

Here, $h_{elec}(l)$ represents the response of the volume conduction environment to a unitary pole located at the 3D coordinate $l$. This response was determined numerically using the Finite Element Method (FEM) [25] by solving a set of boundary conditions.

The potential distribution $V(l)$ within the biological tissues can be modeled from the principles of electric current conservation [26] as:

$$\nabla \cdot \sigma_b(l) \nabla V(l) = 0 \tag{8}$$

where $\sigma_b(l)$ is the conductivity at $l$. No current flow was allowed at the points in the interface between the skin and the air as in:

$$\sigma_b(l) \nabla V(l) \cdot n(l) = 0 \tag{9}$$

where n(l) is a unitary normal vector directed outwards on the domain boundary. Elbow and shoulder points where the model was truncated corresponding with lowest and highest values of Z-axis were set as ground ($V(l) = 0$).

Making use of the principle of reciprocity, the current $J_{Q_0}$ produced by the unitary pole $Q_0$ was imposed at the electrode location as in:

$$\sigma_b(l) \nabla V(l) \cdot n(l) = J_{Q_0} \tag{10}$$

All boundary conditions were implemented in Matlab including the weak form of the Laplace equation for the electromagnetic problem (see Eq 8). Conductivities were established based on the volume conduction grid, allowing for precise and flexible delineation of conductive regions with arbitrary shapes. By solving the electric potential distribution at all points in space using FEM, the response function $h_{elec}(l)$ was obtained for each electrode $elec$ (see S1 Fig from supplementary material for a 3D representation of the response function computed for the 6 monopolar electrodes simulated in this work).

From an electrical perspective, the volume conductor environment can be considered linear. Therefore, the total voltage $V_{elec}(t)$ produced by a fiber at a surface location $elec$ at time instant $t$ can be described as the sum of the voltages $V_{elec}^Q(t)$ produced by the $x$ active poles at that instant:

$$V_{elec}(t) = \sum_{p=1}^{x} V_{elec}^Q(t) \tag{11}$$

## Electrodes

Each electrode serves as a reference point in 3D space to calculate the distance between muscle fiber potential poles and the location of sEMG measurements. The arrangement of the electrodes depends on the simulation requirements. In this study, the following methodology to position a grid of electrodes in a realistic distribution was used:

A cylinder with a radius (R) larger than the volume conduction environment was used as reference location. A grid of MxN electrodes was placed in cylindrical coordinates ($r = R, \theta_e, z_e$), where $\theta_e$ and $z_e$ represent evenly spaced coordinates $\theta$ and $z$ of electrode $e$ (M divisions in $\theta$ and N divisions in $z$). For each electrode, a line segment connecting it to the centroid of the volume conduction environment was computed. The intersection point of this line segment with the volume conduction environment was determined and electrodes were positioned at the closest intersection point on the surface (see 'Electrode Placement' in Fig 1B for a visual representation of this method).

Although the inter-electrode distance was initially constant in the cylindrical arrangement, this process resulted in irregularities in the inter-electrode distance due to the non-uniformity of the skin surface, thereby simulating a more realistic electrode distribution.

## Motor neuron

Motor neurons are implemented as MATLAB objects and serve three main functions: a) grouping muscle fiber objects into motor units, b) releasing neurotransmitters at the neuromuscular junctions of their associated fibers, and c) obtaining the combined force and voltage profiles produced by the active fibers of the motor unit.

**Grouping of muscle fibers.** Each motor neuron object maintains references to all the muscle fiber objects it innervates. This allows customization of motor unit sizes by specifying the number of fibers innervated by each unit (with the requirement that the total number of muscle fibers equals the product of the number of motor neurons and the number of fibers innervated by each neuron). Additionally, there are pseudo-customizable properties for the resulting motor units, such as the dispersion area and the motor unit type. However, the assignment of these properties to motor units is constrained by the pre-established physical distribution of the muscle fibers (this will be further clarified in Configuration of simulation parameters section).

**Neurotransmitters release.** To accurately simulate the motor neuron mechanism responsible for muscle fiber recruitment within the motor unit domain, a mathematical model is employed to describe the profile of released acetylcholine ($ACh$) for a single impulse. The formula is as follows:

$$ACh(t) = A_m \cdot \left(1 - e^{-k_r(t-t_p)}\right) \cdot e^{-k_d(t-t_p)} \tag{12}$$

where, $A_m$ represents the maximum concentration of acetylcholine released, while $k_r$ and $k_d$ are the release and degradation rates, respectively. $t_p$ is the time at which the impulse reach the neuromuscular junction (see Fig 1C1).

By modifying these parameters, the acetylcholine profile for a single impulse from the motor neuron can be customized. Furthermore, the temporal summation of acetylcholine profiles resulting from multiple impulses with varying firing rates allows for the achievement of different acetylcholine peaks at the neuromuscular junction, effectively simulating the motor neuron's ability to modulate recruitment within its pool of muscle fibers (see Fig 1C2).

**Compute motor unit force and voltage profiles.** The motor neuron object is tasked with calculating the combined voltage ($V^e_{mu}(t)$) and force ($F_{mu}(t)$) exclusively from the $n$ innervated muscle fibers within the motor unit at each time instant $t$ as in:

$$V^e_{mu}(t) = \sum_{f=1}^{n} V^e_f(t) \tag{13}$$

$$F_{mu}(t) = \sum_{f=1}^{n} F_f(t) \tag{14}$$

where $V^e_f(t)$ is the voltage produced by all the active poles of the fiber $f$ at the time $t$ in the electrode position $e$ (see Eq 11) and $F_f(t)$ is the force profile of the fiber $f$ at the instant $t$ (see Eq 6)

## Motor control system

The motor control system encompasses a wide array of tasks, including goal-oriented behavior, muscle fiber recruitment, motion planning, and muscle coordination. This study focuses specifically on developing a framework for simulating muscle contraction. Therefore, the motor control system tasks considered here are limited to those directly related to muscle contraction. In this context, the key tasks of the motor control system involve: a) Selecting

and recruiting motor units through the activation of specific motor neurons and b) Determining the firing rates at each time instant to recruit an adequate number of muscle fibers, thus meeting the requirements for the desired contraction force (see Fig 1D).

Once the control strategy is chosen the simulated sEMG and force profile can be computed as:

$$sEMG^e_{MonoPolar}(t) = \sum_{mu=1}^{n} V^e_{mu}(t) \tag{15}$$

$$sEMG^{e_x-e_y}_{BiPolar}(t) = \sum_{mu=1}^{n} \left[ V^{e_x}_{mu}(t) - V^{e_y}_{mu}(t) \right] \tag{16}$$

$$F_{cp}(t) = \sum_{mu=1}^{n} F_{mu}(t) \tag{17}$$

where $sEMG^e_{MonoPolar}(t)$ is the sEMG signal recorded by the electrode $e$ and $sEMG^{e_x-e_y}_{BiPolar}(t)$ is the sEMG signal recorded by the bipolar arrangement between electrodes $e_x$ and $e_y$. Both are computed from $V^e_{mu}(t)$ (see Eq 13) which is the voltage produced by the $mu$ motor neurons for $mu = 1, 2, ..., n$ recorded by the electrode $e$. Finally, $F_{cp}(t)$ represents the total force contraction profile computed by the addition of the force profiles of all the motor units involved in the contraction ($F_{mu}(t)$, see Eq 14).

This system offers flexibility by allowing users to define their own strategies for motor unit recruitment and fiber activation. This empowers researchers to explore previously challenging motor recruitment mechanisms and adapt the module's behavior to their specific movement goals. By leveraging the comprehensive parameters and functions of individual muscle fibers in the model, users can optimize the control strategy. This approach facilitates efficient recruitment of motor units and fibers based on the desired force, velocity, and contraction patterns. The motor control system module's behavior is determined by the choices made by the experimenter, highlighting its adaptability to diverse simulation scenarios. Configuration of simulation parameters section describes the details of the motor control strategy logic implemented for the muscle contractions simulated in this work.

## Model considerations

The presented model is a modular system that emulates the behavior of the basic components of the muscular contraction system. Each element within the system has independent functions and parameters that can be modified without affecting the overall functionality. Care should be taken only with shared functions among modules. For instance, if a different method of muscle fiber innervation is desired, both the neurotransmitters emission function in the motor neuron and the reception and innervation capability in the muscle fiber need to be adjusted.

Furthermore, the system offers high configurability to accommodate a wide range of muscular contractions. This flexibility allows for the inclusion of additional functionalities in the future. For example, the current system does not consider the physical contraction of muscle fibers, meaning their spatial position remains fixed during innervation. However, incorporating such functionality (albeit computationally demanding) could enhance the realism of simulating isotonic movements. It is also possible to introduce extra variables and functions to model other fiber behaviors, such as temporary metabolic changes or fatigue.

## Configuration of simulation parameters

To validate the system, a simulation of a human biceps comprising tens of thousands of muscle fibers will be conducted, necessitating the establishment of a logical framework for configuring simulation parameters. This section will be divided into three blocks, detailing the logic employed to set the parameters for the simulation environment (arm section and electrodes), the arrangement and distribution of fibers and motor neurons within the biceps, and the muscle contraction strategy.

Table 1 presents the comprehensive set of parameters used for the ensuing simulations together with the literature used as reference.

**Muscle environment.** The initial parameters to define are the boundaries of the simulated arm section. The X, Y, and Z ranges were determined based on MRI scans of the upper arm obtained from the public tomologic atlas from the Visible Human Project (Table 1 - Arm Dimensions, [28]).

To create the finite-point grid representing the volume distribution of four biological tissues, a total of 37 scans were utilized. These scans had an inter-scan distance of 4 mm in the longitudinal direction. Fig 2A provides a visual representation of the tissue layers (cancellous bone in white, cortical bone in blue, muscle in red, and fat/skin in grey). The electrical conductivity properties of each tissue are presented in Table 1 - Arm Dimensions.

For the electrode setup, a matrix of 2x3 electrodes was employed with a 1.5 cm distance between electrodes. The electrode grid was centered at 30% of the z-axis to ensure consistency with the location of the neuro-muscular junction in the bicep muscle according to the Surface Electromyography for the Non-Invasive Assessment of Muscles (Seniam) Project guidelines [43]. Furthermore, the electrodes were adjusted to the skin using the methodology described in Electrode section.

**Properties and distribution of fibers and neurons.** After selecting the number of bicep fibers, denoted as $n_{Fib}$, the relative position and size of the biceps muscle within the simulated arm were determined using cross-sectional areas from 7 MRI scans [28]. Each bicep area in the scans was divided into $n_{Fib}$ equally spaced points.

To connect the fiber points between consecutive MRI scan layers, the following process was employed: each scan was vertically or horizontally divided (depending on the dimensions of the largest area) into two areas with an equal number of points. This process was repeated iteratively for each paired layers until reaching single point areas. Once this was achieved, each point in the first layer areas was associated with a corresponding single points in the second layer areas. This process was repeated for all seven scans, resulting in a collection of seven 3D points ($Fib_{pos}$) for each muscle fiber (refer to Fig 2B).

The motor neuron number ($MN_{num}$) and motor unit sizes ($MN_{size}$) were initially selected from a Gaussian distribution with given average and standard deviation values (see Table 1 - Motor Unit Parameters). The resulting number of fibers was adjusted by adding or removing fibers from randomly chosen motor units until it matched the previously determined value of $n_{FIB}$. Acetylcholine profiles generated by motor neurons (refer to Eq 12) were defined based on fixed values of $A_m$, $k_r$, and $k_d$ (see Table 1 - Acetylcholine Profile).

The following approach was employed to determine the classification of each muscle fiber into types I, IIa, and IIx. Initially, the desired proportions of each muscle fiber type (**%TI**, **%TIIa**, **%TIIx**) were established. Subsequently, types were randomly assigned to each fiber. Then, three Gaussian distributions were defined, with the same standard deviation but different average values evenly distributed across the range of motor unit sizes. These Gaussian distributions were utilized in an iterative process to transform the muscle fiber types from the initial random distribution to achieve the desired percentages (**%TI**, **%TIIa**, **%TIIx**). The

**Table 1. Simulation parameters configuration.**

| Parameters | | | | References |
|---|---|---|---|---|
| Muscle Environment | | | | |
| Arm Dimensions | | | | |
| X | [-50 50] mm | | | [27,28] |
| Y | [-50 50] mm | | | [27,28] |
| Z | [0 150] mm | | | [27,28] |
| Tissue Conductivity | | | | |
| Cancellous Bone | 7.56e-2 S/m | | | [29] |
| Cortical Bone | 2.00e-2 S/m | | | [29] |
| Muscle | 0.30 S/m | | | [29,30] |
| Fat/Skin | 4.07e-2 S/m | | | [29] |
| Electrodes | | | | |
| Inter-electrode dist (z-axis) : | $\sim$ 1.5 cm | | | |
| Biceps Parameters | | | | |
| Range of Fibers | 200k f | | | [31–33] |
| Motor Unit Parameters | | | | |
| Size and Dispersion | | | | |
| Motor Unit size (avg $\pm$ std) | 300 $\pm$ 100 f/mu | | | [33] |
| Motor Unit number (avg $\pm$ std) | 1000 $\pm$ 250 mu | | | [33] |
| $R$ range | 4 - 40 % of muscle cross section area | | | - |
| $\lambda$ range | 20 - 50 % of $R$ | | | - |
| Acetylcholine Profile | | | | |
| $A_m$ | 1 mM | | | - |
| $k_r$ | 0.45 $s^{-1}$ | | | - |
| $k_d$ | 30 $s^{-1}$ | | | - |
| Muscle Fiber Parameters | | | | |
| From Fiber Type | | | | |
| | Type I | Type IIa | Type IIx | |
| $r$ gaussian (avg $\pm$ std) | 20 $\pm$ 5 $\mu m$ | 27.5 $\pm$ 7.5 $\mu m$ | 30 $\pm$ 10 $\mu m$ | [34,35] |
| $\sigma_{ic}$ gaussian (avg $\pm$ std) | 0.35 $\pm$ 0.25 $S/m$ | 0.5 $\pm$ 0.5 $S/m$ | 0.7 $\pm$ 0.5 $S/m$ | [36–38] |
| $thr$ gaussian (avg $\pm$ std) | 0.45 $\pm$ 0.15 $mM$ | 0.6 $\pm$ 0.2 $mM$ | 0.9 $\pm$ 0.3 $mM$ | - |
| $F_{base}$ gaussian (avg $\pm$ std) | 65 $\pm$ 15 $mN$ | 75 $\pm$ 20 $mN$ | 100 $\pm$ 25 $mN$ | [35,39,40] |
| $\tau_{UP}$ gaussian (avg $\pm$ std) | 37 $\pm$ 12 $ms$ | 15 $\pm$ 5 $ms$ | 7 $\pm$ 3 $ms$ | [35,39,40] |
| $\tau_{DOWN}$ gaussian (avg $\pm$ std) | 50 $\pm$ 14 $ms$ | 20 $\pm$ 7 $ms$ | 10 $\pm$ 4 $ms$ | [35,39,40] |
| From Other Data | | | | |
| $fib_{pos}$ | From scans | | | [28] |
| $cv$ range | 3-5 m/s ($\propto r \cdot \sigma_{ic}$) | | | [41] |
| $nmj_{pos}$ gaussian (avg + std) | 40 $\pm$ 5 % (from upper tendon) | | | [42] |
| Fix Parameters | | | | |
| $A$ | 96 mV | | | [6] |
| $B$ | 90 mV | | | [6] |
| $\alpha$ | 550 | | | - |
| $C$ | 1500 | | | - |

Gaussian distributions with the lowest, medium, and highest average values were employed to select small, medium, and large motor unit fibers for conversion into type I, IIa, and IIx, respectively, until the desired type distribution was achieved. This methodology enables the presence of fibers of any type within each motor unit, while promoting the tendency for fibers to be grouped from slowest to fastest based on motor unit size [44].

Constant parameters were used for all fibers in the computation of the voltage distribution ($V_m(z)$), including the amplitude of the action potential (**A**), resting membrane potential (**B**), scale factor (**$\alpha$**) and proportionality constat (**C**) (see Table 1 - Fixed Parameters). Fiber radius (**r**), intracellular conductivity (**$\sigma_{ic}$**), ACh threshold (**thr**), and force properties (**$F_{base}$**, **$\tau_{Up}$**,

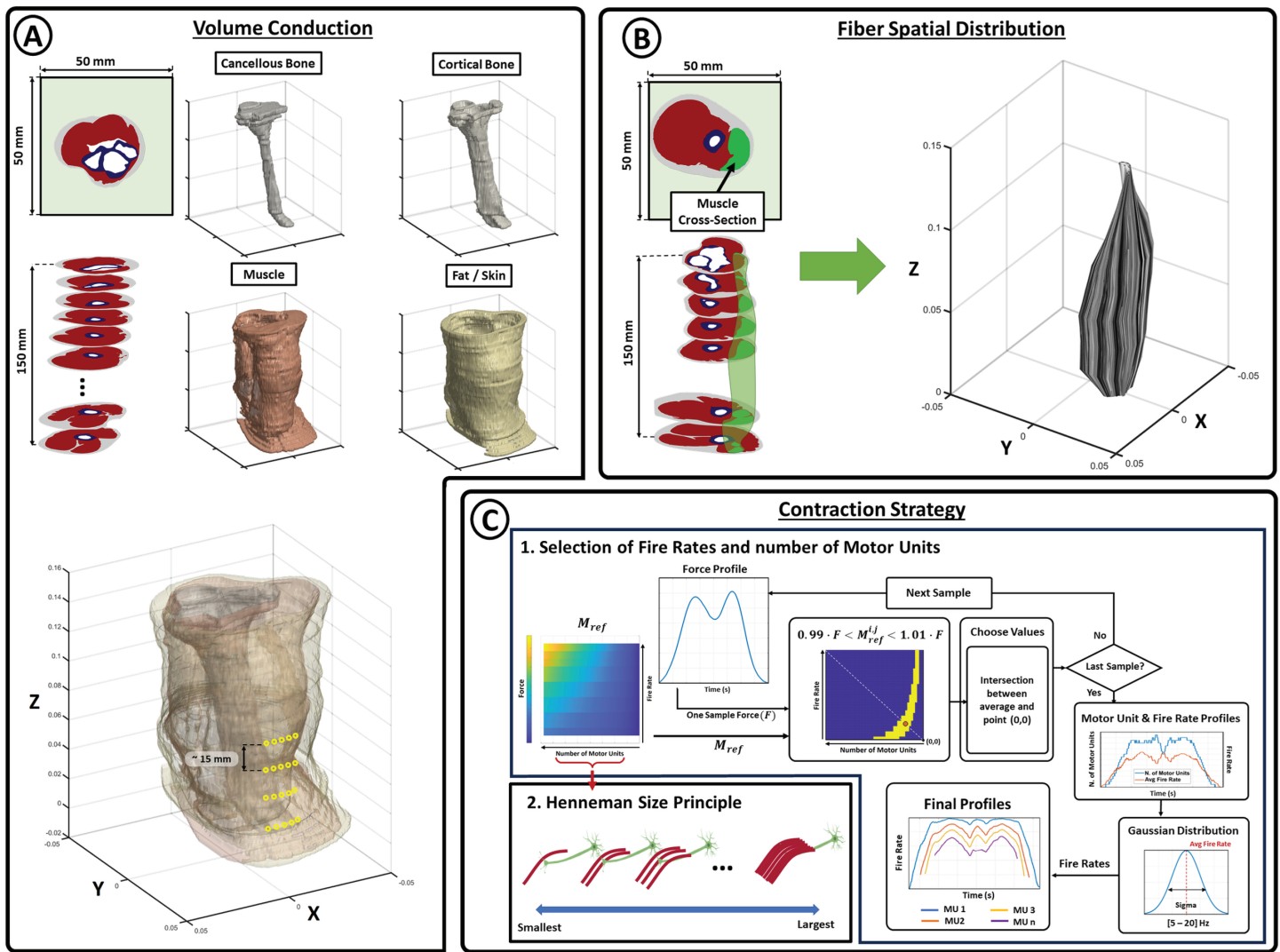

**Fig 2. Model configuration.** Methodology for the definition of the volume conduction environment (A), fiber spatial distribution (B) and muscle contraction strategy (C).

$\tau_{Down}$) were assigned based on Gaussian distributions, with the mean and standard deviations related to the muscle fiber type (see Table 1 - Fiber Properties). The position of the neuro-muscular junction ($\textbf{nmj}_{pos}$) were also assigned based on a Gaussian distribution (see Table 1 - Fiber Properties). They were initially assigned as a percentage of the arm length, and the final x, y, z coordinates were computed as the intersection between the fiber position ($\textbf{Fib}_{pos}$) and the z-plane at the specific length. Fiber conduction velocity ($\textbf{cv}$) was assigned within a range of 3-5 m/s, proportional to the product of the fiber ratio and intracellular conductivity ($\textbf{r} \cdot \boldsymbol{\sigma}_{ic}$) [45].

Finally, the fibers of each motor unit needed to be aligned with the spatial distribution ($\textbf{Fib}_{pos}$) derived from the MRI scans. For this purpose, first motor unit fibers were dispersed using the following formula:

$$Disp_{mu} = R \cdot Gaus_{2D}(0, \lambda) \qquad (18)$$

Here, $Gaus_{2D}(0, \lambda)$ generates the x and y coordinates of a 2D Gaussian distribution centered at 0, with a standard deviation of $\lambda$. The values of $\lambda$ and $R$ indicate the extent and density of fiber dispersion for each motor unit. Both values were obtained from Gaussian distributions, with their means and standard deviations (refer to Table 1 – Motor Unit Parameters) proportional to the size of the motor unit. The values of $R$ and $\lambda$ were expressed as percentages of the bicep cross-sectional area.

The purpose of this method was to incorporate a tendency for smaller motor units to exhibit smaller radii and lower fiber densities, while larger motor units displayed larger radii and higher fiber densities. Subsequently, the motor units were assigned to the reference points (***Fib_pos***), following these steps: Initially, a motor unit was randomly selected, and re-centered into the x and y coordinates of an unassigned fiber (also randomly selected). Subsequently, each fiber from the dispersion was individually assigned to the closest unassigned fiber within the cross-sectional area. This process was repeated for each motor unit until all fibers were associated with their corresponding ***Fib_pos*** (see S3 Video from supplementary materials for a visual representation of this process). This method allowed for flexibility in choosing the initial dispersion properties of the motor units and enabled adjustments to these properties until they matched the actual spatial distribution of fibers. Also, final fiber distribution properties aligned with those reported in literature [46], where each motor unit area, while well defined, was also shared and overlapped with other motor units.

**Contraction strategies.** The present study utilizes time-force profiles as a basis for generating muscle contractions. The motor control system adjusts the recruitment and firing rates of motor neurons by comparing the desired time-force profile with the resulting force obtained from the cumulative force profiles of all active muscle fibers. To achieve this goal, the force response of each motor neuron was computed for each fire rate from 8 to 42 Hz [8]. For each frequency and neuron, the average value of the generated force was recorded in an *NxM* matrix where N was the number of motor units and M was the number of available frequencies. Using this matrix, a second *NxM* matrix ($M_{ref}$) was defined to measure a cumulative force produced by each number of neurons and fire rate by following the Henneman's size principle [47] (i.e. motor units are recruited from smallest to largest, see Fig 2-C1). Therefore the position $N = 5, M = 10$ contains the cumulative force of the 5 smallest neurons activated with a fire rate of 10 Hz.

Fig 2-C1 shows how this second matrix was used to choose the number of neurons and activation profiles on each time instant. First, the reference force profile was evaluated sample by sample. The force of each sample ($f_{samp}$) was set as goal and the required number of neurons $n$ and fire rate $m$ were chosen from a set of values from $M_{ref}$ that satisfies $0.99 * f_{samp} > M_{ref}^{n,m} > 1.01 * f_{samp}$. Fig 2C1 shows how $n$ and $m$ where chosen as the middle point from the intersection between the point 0,0 and the point $MU_{max}, FR_{max}$, where $MU_{max}$ and $FR_{max}$ are the maximum number of Motor Units and Fire Rate respectively. This process was repeated for each sample resulting in two profiles indicating the temporal evolution of the recruited motor units and their average activation frequency. This algorithm assures a progressive increase of both parameters with the desired force. Moreover, to simulate the progressive increase of the fire rate of newly recruited motor units and to fit the experimental results obtained in literature [48], firing rates were assigned from a Gaussian distribution with a mean frequency given by the average fire rate profile and a standard deviation of 10 Hz [49], linearly related to the number of recruited neurons (see S4 Video from supplementary materials for an example of the recruitment of motor units and fire rates for a given force profile).

Once the profiles were determined, the propagation of muscle fiber potentials was simulated for all active fibers, and the surface electromyography (sEMG) signal was computed according to Eqs 15 and 16.

## Parameter considerations

In configuring the parameters of the muscle contraction, it is important to acknowledge the inherent challenges posed by inter-subject variabilities and the limited availability of comprehensive literature. Due to the complex nature of individual physiological responses and the lack of sufficient data addressing specific parameter values, certain assumptions based on general information derived from existing literature were made in this work. These assumptions were necessary to distribute the parameter values within the model and account for the known variabilities.

For example, when distributing the properties of individual muscle fibers, certain parameters such as radius, intracellular conductivity, maximum force, and force rise and decay times were assigned. While literature does not provide the exact values for each muscle fiber type, it is generally accepted that faster fibers tend to have larger radius, higher intracellular conductivity, greater maximum force, and shorter force rise and decay times. To capture these relationships, ranges for each parameter were derived from existing literature and organized them into meaningful groups corresponding to each fiber type. Additionally, fiber activation thresholds for the fibers and acetylcholine release profiles were defined in a way that recruiting slower or faster fibers is only achievable by increasing the motor neuron firing rate, which aligns with widely accepted principles. These assumptions aim to capture the essential characteristics of muscle behavior while acknowledging the complexities and limited data availability associated with precise parameter estimation. It is worth noting that while these approximations provide a valuable foundation for analysis, further investigations and additional data collection are needed to optimize this parametrization.

## Simulation

sEMG signals were simulated under three different parameter configurations, all of which employed the same spatial distribution of bicep area and biological tissues. The primary variation among these configurations was the distribution of fiber types. Configurations 1 represents the average fiber type distribution of the bicep brachii for both young and elderly individuals, as documented in Monemi et al. (1998) [50]. Configurations 2 and 3 are predefined examples with a dominance of slow and fast-twitch fibers, respectively (refer to Table 2). The variables "S1 Files", available as Supplementary materials contain all motor unit and individual fiber parameters. For each parameter configuration, both short-term and long-term contractions were simulated at a sampling frequency of 2000 Hz.

As introduced in Muscle fiber section, the current model necessitates temporal force profiles to guide the motor control system in determining the required number of motor units and firing rates to achieve the specified force output. In this simulation, real force profiles recorded during two types of movements were utilized: periodic isotonic contractions and sustained isometric contractions. Before their application to the model, real force profiles were fitted to each model characteristic. For this, force profiles were normalized according to

**Table 2. Models' fiber type distribution.**

| | Fiber Type (%) | | | |
|---|---|---|---|---|
| Configuration | TI | TIIa | TIIx | Data (Supl. Materials) |
| 1 | 50 | 28 | 22 | conf_1.mat |
| 2 | 70 | 15 | 15 | conf_2.mat |
| 3 | 30 | 20 | 50 | conf_3.mat |

the maximum voluntarily contraction (MVC) of each participant and proportionally adjusted to the MVC of the model under evaluation. The following section outlines the experimental details for obtaining these reference profiles.

## Experiment and data

Fig 3 depicts the experimental setup used for measuring force profiles and the corresponding sEMG signals during elbow flexion tasks.

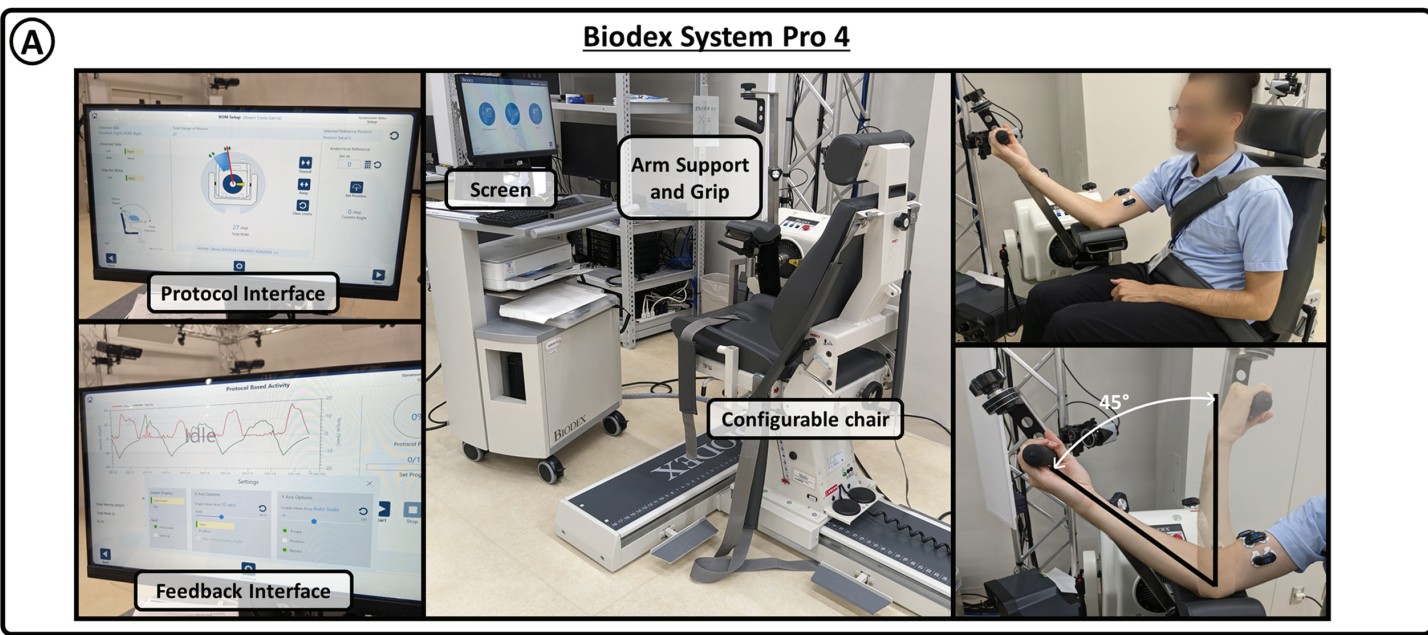

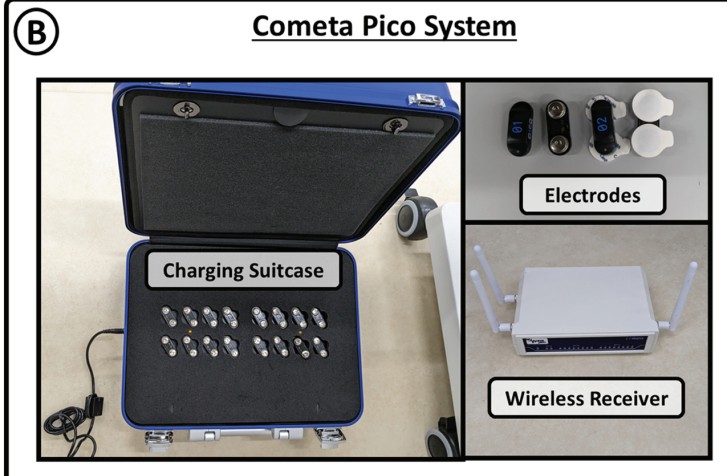

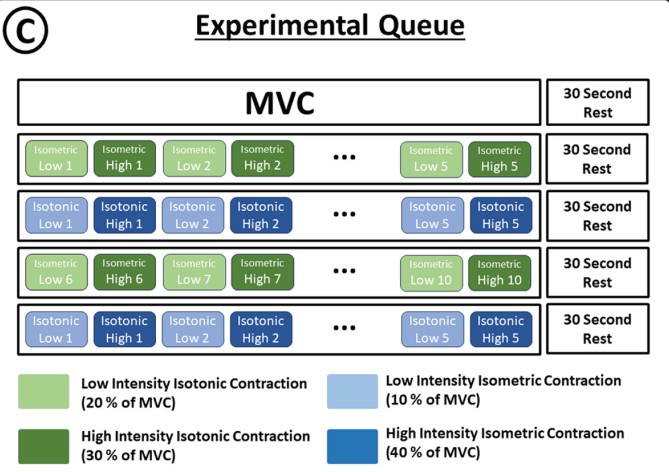

**Fig 3. Experimental set-up and protocol.** A) Biodex System Pro 4: The central image provides an overview of the system composed of a chair, the dynamometer, and the computer processing and displaying the data. The images on the left depict the task configuration screen and the feedback generated during the motion. The images on the right show the dynamometer adapter, allowing for elbow flexion and extension tasks, and illustrate the predetermined range of motion set for the experiments in this study. B) Cometa Pico System: sEMG signal measurement system consisting of wireless bipolar wet electrodes and the receiver used for data storage in a computer. C) Experimental Queue: In the initial trial, participants were instructed to exert maximum voluntary contractions. Subsequent trials involved alternating high and low force level isotonic and isometric contractions according to the image. A 30-second rest period concludes each trial.

**Force profiles.** Force profiles were measured using the Biodex System Pro 4 (Biodex Medical Systems, Inc.), a device that allows torque measurement with a sampling frequency of 100 Hz via a dynamometer (See Fig 3A). The system enables the generation of experimental protocols involving the limbs and various types of muscle contractions while providing real-time feedback on the torque profile.

**sEMG measurement.** sEMG signals were simultaneously recorded using an arrangement of three bipolar electrodes positioned on the central and lateral parts of the biceps brachii, following the guidelines of the Surface EMG for non-invasive assessment of muscles (SENIAM) project [43] (see Fig 3A–B). The Cometa Pico EMG measurement system (Cometa Systems, Inc.) was employed to digitize the signals at a sampling rate of 2000 Hz.

**Experimental protocol.** Each participant was instructed to sit in the chair integrated into the Biodex system. The chair and dynamometer position were adjusted to facilitate comfortable elbow flexion and extension movements within a maximum range of 45 degrees, spanning from the point of maximum extension to a position perpendicular to the ground (see Fig 3A - right image). The participant's body was secured to the chair with a strap, and an elbow support was used to minimize any movement in the biceps region, except for natural muscle contraction during tasks. These precautions were taken to prevent low-frequency motion artifacts from affecting the sEMG signals. Additionally, the experimental room was isolated to minimize the coupling of white Gaussian noise, and wireless sEMG electrodes were used to avoid power line interference. The experiment consisted of two tasks, each with two intensity levels.

The first task involved isotonic biceps flexion movements in which the user was instructed to perform flexions between the points of maximum extension and maximum flexion one at a time. The participant was asked to execute flexions at a comfortable speed. After each flexion, the grip attachment was returned to the maximum extension position by the experimenter to minimize participant's muscle activation during extension.

The second task entailed 5-second isometric contractions from the previously configured maximum extension position. During this task, the dynamometer prevented rotation but continued to provide feedback of the torque force generated by the participant.

Each of these tasks was repeated twice with different force constraints. In the isotonic mode, the Biodex system only allowed the dynamometer to rotate when a certain force threshold was reached. For this task, the established force levels were 20% and 30% of the MVC. For the isometric movements, the force constraints were 10% and 40% of the MVC. In this case, the user was instructed to reach the desired force level using the real-time torque displayed on the Biodex system screen as a reference.

At the beginning of each experimental session, participant's MVC was measured. For each task and force level, there were 10 repetitions of the movement according to the protocol shown in Fig 3C, with 30-second breaks between tasks. Balance between higher and lower intensity contractions was included in the design of the experimental queue in order to avoid fatigue related bias. To familiarize users with the Biodex system, they were allowed to practice both isotonic and isometric movement protocols before each experimental session.

**Data curation.** For each user and task, torque and muscle individual activations were extracted. Torque data was resampled to 2000 Hz to match the sampling frequency of both the sEMG data recorded by the Cometa System and the model-based simulated sEMG (the variables 'S2 Files' in the supplementary materials compile these individual activations). The spectral distribution of the measured sEMG signals was carefully evaluated to ensure the absence of common artifacts such as white Gaussian noise in the higher frequencies, power line interference at 50/60 Hz, and motion artifacts in the lower frequencies [51].

**Participants.** A total of 5 participants took part in the experiment, all of them males, with ages ranging from 33 to 51 (42.8 ± 6.5) years. All participants were right-handed and had no medical history related to motor disorders. Prior to the experiment, all participants were informed about the experimental protocol and provided written informed consent in accordance with the Helsinki Declaration. All procedures were performed in compliance with relevant laws and institutional guidelines in accordance with the ethical approval of AIST (Ref Number: 2023-1406, Approval Date: March 13, 2024). Participants were recruited from March 15 to April 15 of 2024.

## Data validation

Fig 4A depicts the procedural framework for simulating and validating surface electromyography (sEMG) signals. In this context, torque data obtained from the experiments detailed in Experiment and data section served as reference force profiles to initiate simulations based on established models, as outlined in Simulation section. These simulations produced force and sEMG signals for each model under consideration.

Subsequently, an in-depth analysis was conducted to assess the similarity between the sEMG signals generated by the model and those acquired through experimental measurements. To accomplish this, four key characteristics of the sEMG signals were compared, as illustrated in Fig 4B.

To evaluate similarity in terms of amplitude and power over time, the Root Mean Square (RMS) metric was computed. Dominant frequency similarity is assessed using the median frequency (MF), defined as the frequency value that equally divides the spectrum (determined

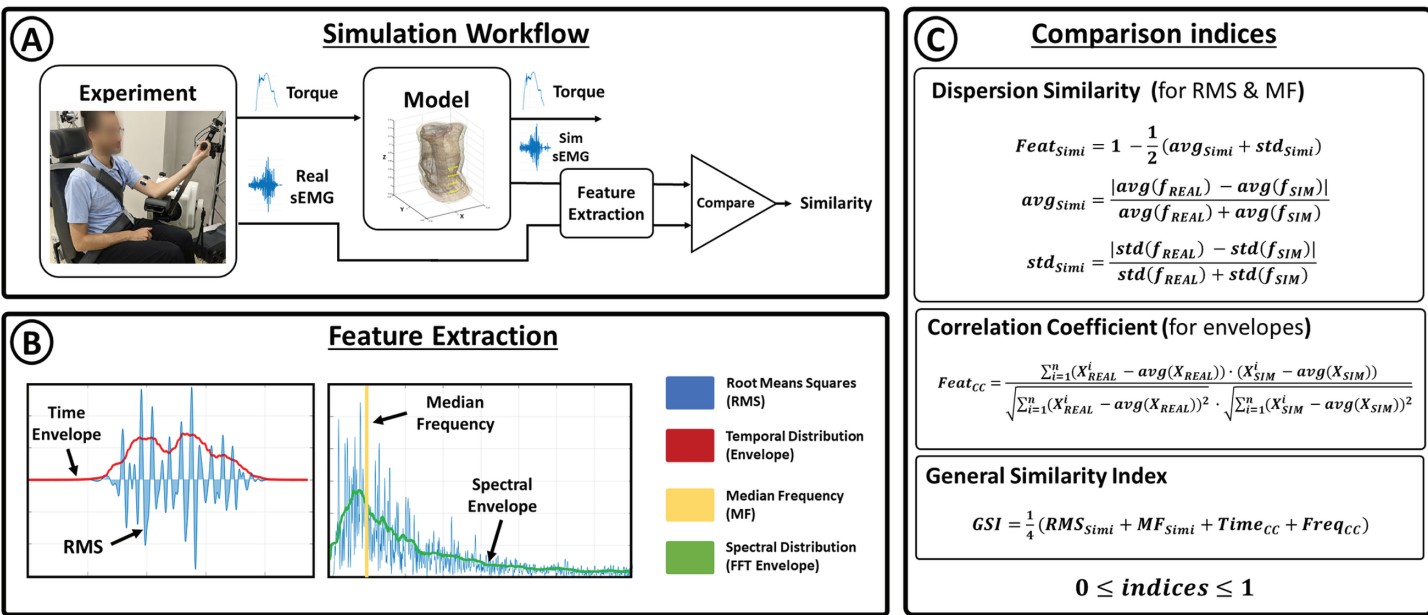

**Fig 4. Data validation**. A) Simulation Workflow: Both sEMG and Torque are measured from participants using the Biodex System Pro 4. Torque data serves as input in the muscle model to generate simulated sEMG signals. The features of simulated and real sEMG are compared to assess their similarities. B) Feature Extraction: Visual representation of the four features extracted from sEMG data. C) Comparison Indices: Mathematical formulation of the indices generated from the extracted features, employed to compare similarities between simulated and real sEMG signals.

through Fast Fourier Transform, FFT). Additionally, the temporal and spectral distributions of the real and simulated sEMG were compared from their envelopes. To obtain these envelopes, the signals undergo rectification and were subjected to a 0.5-second moving average filter for temporal data and a 10-Hz moving average filter for spectral data.

Fig 4C presents the formulation of indices employed for measuring similarity. For RMS and MF, the indices $RMS_{Simi}$ and $MF_{Simi}$ were computed using the following formula:

$$Feat_{Simi} = 1 - \frac{1}{2}(avg_{Simi} + std_{Simi}) \tag{19}$$

Where $avg_{Simi}$ and $std_{Simi}$ represent the similarity on the mean and standard deviation between real and simulated features, defined as:

$$avg_{Simi} = \frac{|avg(f_{REAL}) - avg(f_{SIM})|}{avg(f_{REAL}) + avg(f_{SIM})} \tag{20}$$

$$std_{Simi} = \frac{|std(f_{REAL}) - std(f_{SIM})|}{std(f_{REAL}) + std(f_{SIM})} \tag{21}$$

Here, $f_{REAL}$ and $f_{SIM}$ denote the distributions of a feature (RMS or MF), whether real or simulated, respectively. These distributions encompass all repetitions corresponding to the same participant, task, and intensity.

Additionally, the temporal and spectral envelopes were compared using correlation coefficients, namely $TIME_{CC}$ and $FREQ_{CC}$ for temporal and spectral envelopes (see $FEAT_{CC}$ in Fig 4C) [52].

Where $X^{REAL}$ and $X^{SIM}$ are the vectors comprising $i = 1, 2, ...n$ elements, defining each envelope (temporal or spectral).

Finally, a General Similarity Index (GSI) was defined as the average of the $RMS_{Simi}$, $MF_{Simi}$, $TIME_{CC}$, and $FREQ_{CC}$

$$GSI = \frac{1}{4}(RMS_{Simi} + MF_{Simi} + TIME_{CC} + FREQ_{CC}) \tag{22}$$

It is important to note that all the presented indices are constrained to the range between 0 and 1, where a value of 1 indicates perfect similarity between the compared characteristics, while 0 indicates complete dissimilarity. For the correlation coefficients, which can range between -1 and 1, any value below zero is setted to zero and considered indicative of no correlation. This standardized scale facilitates a straightforward interpretation of the degree of similarity between the simulated and real sEMG signals, providing a meaningful basis for a comparative analysis.

## Results

### Real vs simulated sEMG

The left graphs in Fig 5 illustrate examples of real (blue) and simulated (red) sEMG signals for each movement condition. Additionally, the right graph compares their spectral characteristics. For each frequency ranging from 1 to 500 Hz, a pair of boxplots is presented, depicting the energy distribution for each condition. The use of boxplots facilitates a more transparent visualization of the similarity in energy distribution across frequencies between real and simulated sEMG signals (variables 'S3 Files' in the Supplementary materials contain the set of simulated sEMG corresponding with each real repetition recorded).

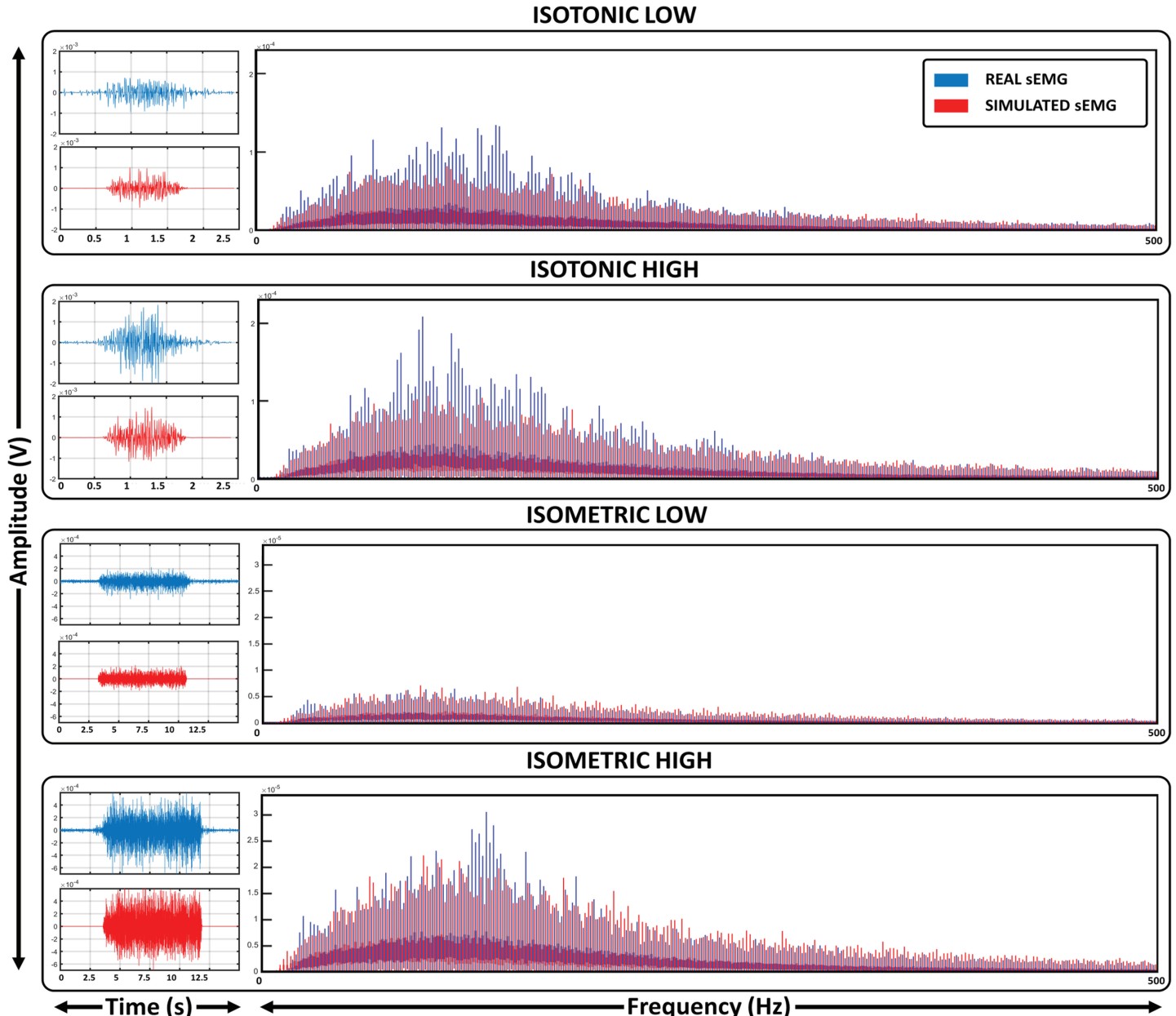

**Fig 5. Temporal and spectral distributions**. The left graphs presents an example of the temporal distribution of real (blue) and simulated (red) sEMG signals for each contraction condition. The right graphs compares their spectral distribution across the entire set of repetitions. The simulated sEMG signals depicted in this figure were computed using Model 1 from Table 1, which includes typical muscle fiber type distributions reported in the literature.

These similarities are further quantified in Fig 6A using the General Similarity Index (GSI) introduced in Eq 22. The blue boxplots represent the GSIs resulting from comparing simulated sEMG signals with real ones. Each GSI included in the boxplot was obtained by comparing datasets from the same user. In contrast, the red boxplots display the GSIs resulting from comparing real signals from one user with those from the rest of the users. To compare both conditions, a Wilcoxon sum-rank test with a 95% confidence interval, modified by a

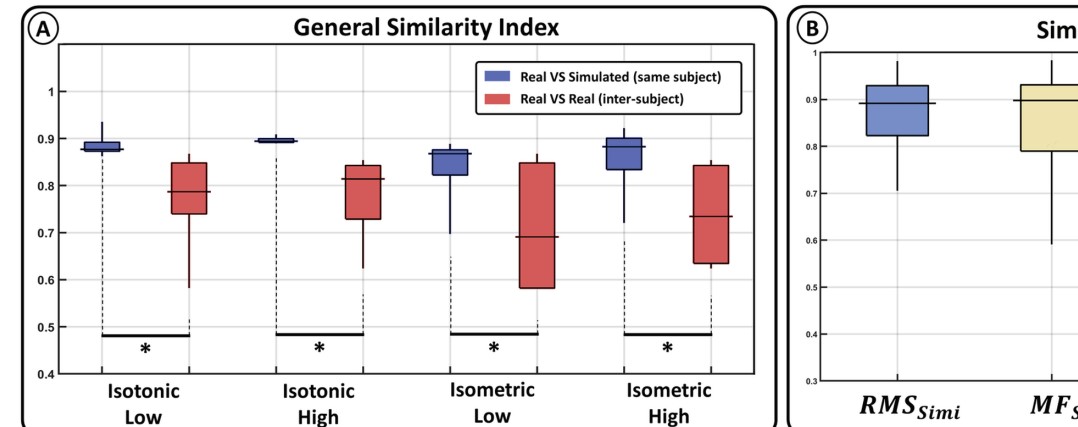
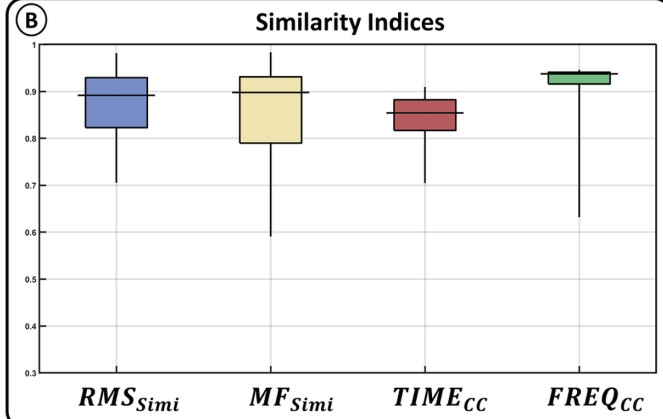

**Fig 6. Similarity indices.** A) General Similarity Index computed between real and simulated sEMG signals (blue) and, between real sEMG signals from one participant and the rest of them (red). B) Comparison between the four similarity indices included into the General Similarity Index when comparing real and simulated sEMG signals.

Bonferroni–Holmes correction to account for multiple comparisons, was employed. Asterisks denote significantly different distribution pairs. These results provide insights into the model's ability to adjust the properties of simulated sEMG to the actual signals recorded for each participant based on their generated force profiles.

Fig 6B illustrates each of the indices comprised within the GSI. The boxplots contain the values obtained for all tasks and users. The figure showcases reconstruction indices averaging above 85% for all evaluated temporal and spectral characteristics. Frequency characteristics ($MF_{Simi}$ and $FREQ_{CC}$) exhibit mean reconstruction values surpassing 90% on average but also demonstrate greater deviations from this mean. In contrast, temporal characteristics ($RMS_{Simi}$ and $TIME_{CC}$) show mean values between 85% and 90% with lower deviations from the mean.

## Fiber type distribution effects

Fig 7 illustrates the effects of employing models with different distributions of muscle fibers (see Table 2). The left graph (Fig 7A) presents the GSI between real and simulated sEMG signals grouped according to models 1, 2, and 3. Additionally, it displays the GSI between real data from different participants. All models generate sEMG signals that exhibit a higher similarity with real signals compared to the similarities observed among real signals from different participants. Notably, Model 1, which adjusts the muscle fiber distribution based on existing literature, produces signals that are more akin to real signals than Models 2 and 3. Statistical analysis also reveals significant differences between the GSI of Model 1 versus Models 2 and 3; however, such differences are not observed when comparing the GSI of Models 2 and 3 against each other.

Furthermore, the graph on the right (Fig 7B) compares the median frequency of real signals with that of sEMG signals simulated using each model. Model 1 emerges as the one capable of generating sEMG signals whose median frequency distribution closely approximates that of real signals. In contrast, Models 2 (slow-twitch dominant) and 3 (fast-twitch dominant) yield distributions of mean frequencies that are lower and higher, respectively, highlighting the propensity of slow and fast-twitch muscle fibers to generate lower and higher frequencies in sEMG signals.

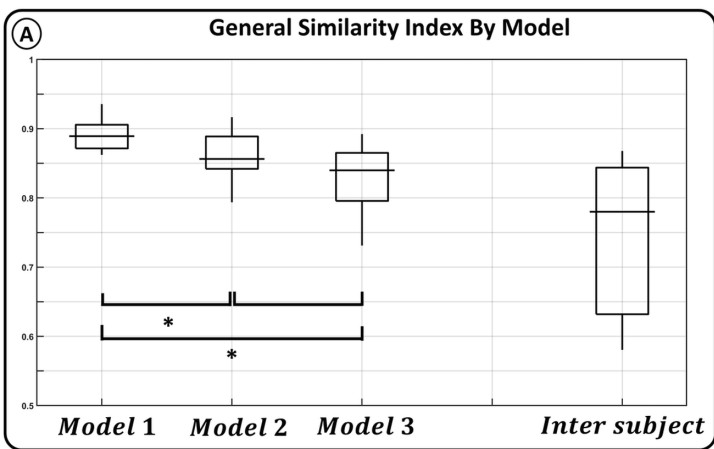
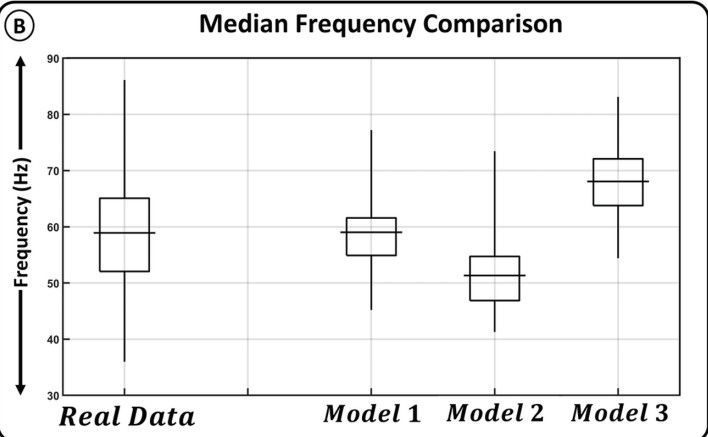

**Fig 7. Effects of muscle fiber distribution.** The left graph (A) displays the General Similarity Index (GSI) between real and simulated sEMG signals for models 1, 2, and 3 (see Table 2), as well as the GSI between real data from different participants. The right graph (B) compares the median frequency of real signals with that of sEMG signals simulated using each model.

## Discussion

### Model novelty

The model proposed in this project breaks down the simulation of sEMG signals into five interconnected elements. Three of these elements model the chain of reactions from the intention of movement to muscle contraction: the motor control system, motor neurons, and muscle fibers. The other two elements, the volume conduction environment and electrodes, model the electromagnetic transformations experienced by membrane potentials, ultimately recorded on the skin as sEMG signals.

Each of these elements contains independent functions that enable a high degree of realism in generating muscle force and sEMG signals.

The motor control system leverages prior knowledge of existing motor units and fibers to design muscle activation strategies capable of generating specific force profiles. This simulates the behavior of muscle contraction in an adult human, where motor learning processes and plasticity have optimized the recruitment of motor neurons and their firing rates [53,54]. The modeled motor neurons offer great flexibility by generating motor units of varying sizes, dispersion, and percentages of each type of muscle fiber [55,56]. Furthermore, the release of acetylcholine profiles, modulated by fire rates, provides an internal mechanism within the motor neuron to recruit more or fewer muscle fibers within its pool, according to the activation thresholds of each fiber [57].

Muscle fibers, on the other hand, are independent units that activate their membrane potential generation mechanisms and force profiles only when they receive sufficient activation from the motor neuron that innervates them. Including a large number of parameters that model fiber properties allows discrimination between fiber groups, which, although not all identical in their parameters, can be classified into supergroups as described in the current research (type I, IIa, and IIx) [58].

The volume conduction environment allows for the segmentation of space around muscle fibers into geometries representative of each biological tissue, providing a basis for addressing the electromagnetic problem of potential propagation produced by an arbitrary pole at a point in space [6]. Additionally, it facilitates the realistic localization of electrodes on the skin,

serving as the reference point for measuring electromagnetic potential variations. Finally, the localization of an array of electrodes enables the simulation of monopolar and bipolar sEMG measurements according to the experimenter's needs.

The fully parametric nature of the proposed model allows it to account for biological variability across populations, such as differences in age, gender, and muscle structure. By adjusting the parameters of each neuromuscular component (e.g., muscle fibers, motor neurons, recruitment strategies), the model can be configured to reflect the specific characteristics of any individual, as long as these values are known. This adaptability makes the framework applicable to personalized scenarios, such as tailored rehabilitation or clinical diagnosis, where individual-specific data is available.

### Simulated vs real sEMG

The simulated sEMG show significant resemblance to real sEMG signals, as evidenced by the spectral distributions presented in Fig 5. Through quantitative analysis (see Fig 6) an average similarity level of approximately 90% is revealed for both temporal and frequency characteristics of the simulated signals. Notably, employing real force profiles yields simulated sEMG signals that closely mirror actual signals, consistently surpassing the resemblance levels observed when comparing inter-participant real sEMG signals for the same task. While various studies have simulated sEMG signals [6,59,60], to the best of the authors' knowledge, this is the first paper to report such detailed similarity levels based on their intrinsic characteristics.

It is worth noting that identical MRI scans were utilized to delineate both biological tissues and muscle shape for all participants. Moreover, the distribution of muscle fiber types in the model remained consistent across all participants. The observed similarity between signals, despite these parameters not being tailored for each user, highlights the unexploited potential of the proposed model to accurately approximate sEMG signals.

Although achieving high similarity between simulated and real data is commendable, in highly parametric models, there will always be a potential for tuning the model to reach a certain degree of similarity. The true significance of our model lies in its grounding in the known physiological and biomechanical processes underlying muscle contractions. By adhering to these principles, our model provides a framework for testing modifications to various elements and assessing their effects on surface electromyography (sEMG) features. This approach enables more nuanced investigations into the relationship between internal parameters and sEMG signals, offering insights into the complex interactions within the neuromuscular system. Additionally, simulation results suggest promising applications for the generation of extensive databases of sEMG signals without relying on human participants. Such an approach circumvents the ethical and technical challenges associated with experimental preparations.

### Model limitations

As is common in any scientific model, this approach to simulating sEMG signals is affected by several limitations that are essential to discuss. In this context, it is useful to distinguish between two types of limitations: those related to the specific implementation of the model and those that affect the model in a more general sense.

**Limitations of the current implementation.** One significant issue is the oversight of spatial muscle fiber contraction during innervation, where factors such as changes in muscle fiber length and skin-muscle displacement might affect the simulation of isotonic contractions. The deviation of maximum values between simulated and real sEMG in the spectral distributions of isotonic tasks shown in Fig 5 could be a result of this shortcoming. Another limitation is

the reliance on joint torques, which involve muscles beyond the biceps. We mitigated this by normalizing force profiles based on participants' maximum voluntary contraction and adjusting them to the simulated bicep's force proportionally; however, the inclusion of a muscle force distribution model could further minimize this effect. In a similar vein, our simplified mechanism of motor unit recruitment and firing rate, while effective, could be enhanced with more complex existing models of motor unit pool organization [49]. Finally, the lack of specific empirical data on neuromuscular parameters also presents a challenge in defining the internal parameters of the model. However, the modular structure presented in this work provides us with the necessary tools to introduce, through future developments, modifications to address these limitations proactively and expanding the realism of the model.

**General limitations.** Due to the complexity of muscular and neurological systems and the need to simulate multiple motor units and muscle fibers, the model requires significant computational resources and extended simulation times. Despite advances in computational technology, this demand remains a barrier to the practical application of the model in certain contexts. However, the authors of this work are confident that future developments in computational technology, coupled with optimization in model algorithms, will allow addressing these limitations and improve the efficiency of the model without compromising its realism.

## Future plans

Apart from the evident interest in enhancing and expanding the functionalities of the presented model, there are more specific use cases planned in the short term. In recent years, several studies have emerged demonstrating the potential for decomposing the spectrum of sEMG signals into two distinct components, whose behavior seems to represent groups of motor unit action potentials with varying frequencies [61,62]. In 2022, the authors of this paper evaluated the evolution of these components during the process of muscle fatigue [63]. The results led to the hypothesis of a potential correlation between these components and the distribution of active muscle fiber types. This remains a topic of ongoing debate [11–13], as it is known that sEMG frequencies are influenced by numerous factors that cannot be simultaneously measured during sEMG recordings. One of the primary motivations for developing this model was to establish a reference framework for linking the characteristics of sEMG signals to both internal neuromuscular parameters, and electromagnetic attenuation and dispersion factors, enabling the study of the influence of each one of them on sEMG frequencies. The results of this study demonstrate significant dependencies on the distribution of muscle fiber types in the median frequency of the simulated signals (see Fig 7). In future works, author will aim to replicate the spectral decomposition reported in real signals using simulated sEMG signals, and then explore correlations between these components and the proportions of active muscle fibers. The ability to estimate the distribution of muscle fiber types using non-invasive methods could lead to significant advancements in digital health, facilitating the development of personalized training strategies for athletes, the early detection of muscular disorders such as sarcopenia and the overall monitoring of muscle condition.

Furthermore, the model holds potential for expansion towards simulating more complex muscle activations. In the future, one of the key objectives will be to implement physical muscle contractions within the model, which would allow it to simulate kinematics. This would involve incorporating skeletal biomechanics and enhancing the motor control system to manage the recruitment of multiple muscle groups in coordinated movements. Such an expansion will also require improvements to the motor control module to manage the temporal recruitment of both synergistic and antagonistic muscles. If these enhancements are achieved, it will

be possible to integrate our model with high-level motor coordination theories, such as muscle synergies, enabling simulations of more intricate movements involving multiple muscle groups.

Additionally, advancements in artificial intelligence and machine learning offer exciting opportunities for exploring and optimizing the extensive parameter space of our model. By leveraging these technologies, this approach has the potential to shed light on the complex interactions between sEMG signals, muscle fibers, motor neurons, and higher motor control strategies, thereby paving the way for new discoveries in the field of human motion.

## Conclusion

In this study, we have developed an advanced computational model for simulating sEMG signals that offers significant improvements in accuracy and flexibility. By focusing on individual muscle fibers and incorporating comprehensive physiological and anatomical properties, our model can realistically simulate sEMG signals from any force profile. Our findings reveal that the model can replicate the temporal and spectral characteristics of real sEMG signals with approximately 90% accuracy, demonstrating its robustness. This high level of fidelity enables a deeper investigation into the relationships between sEMG signal characteristics and internal neuromuscular parameters, providing valuable insights into muscle physiology and motor control mechanisms. Ultimately, our model establishes a solid framework for simulating realistic sEMG signals under various neuromuscular conditions. This work has the potential to advance our understanding of human motor control, aid in the development of extensive sEMG databases, and foster innovations in non-invasive muscle health assessment and personalized training strategies.

## Acknowledgments

We would like to extend our sincere gratitude to all the participants of our experiments for their contributions. We also thank the Hokuriku Center of AIST for providing the excellent facilities that made our experiments possible. Additionally, we appreciate the support from BiodexRehab, whose guidance on utilizing the Biodex System Pro was instrumental in our research.

## Author contributions

**Conceptualization:** Alvaro Costa-Garcia, Shingo Shimoda, Akihiko Murai.

**Data curation:** Alvaro Costa-Garcia, Akihiko Murai.

**Formal analysis:** Alvaro Costa-Garcia, Shingo Shimoda.

**Funding acquisition:** Shingo Shimoda, Akihiko Murai.

**Investigation:** Alvaro Costa-Garcia.

**Methodology:** Alvaro Costa-Garcia, Shingo Shimoda.

**Project administration:** Shingo Shimoda, Akihiko Murai.

**Resources:** Alvaro Costa-Garcia, Akihiko Murai.

**Software:** Alvaro Costa-Garcia.

**Supervision:** Alvaro Costa-Garcia, Shingo Shimoda, Akihiko Murai.

**Validation:** Alvaro Costa-Garcia.

**Visualization:** Alvaro Costa-Garcia.

**Writing – original draft:** Alvaro Costa-Garcia, Shingo Shimoda.

**Writing – review & editing:** Alvaro Costa-Garcia, Shingo Shimoda, Akihiko Murai.

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
