## [Decision Letter · Decision Letter 0]

4 Oct 2024

PONE-D-24-29843Tailoring Neuromuscular Dynamics: A Modeling Framework for Realistic sEMG SimulationPLOS ONE

Dear Dr. Costa Garcia,

Thank you for submitting your manuscript to PLOS ONE. After careful consideration, we feel that it has merit but does not fully meet PLOS ONE’s publication criteria as it currently stands. Therefore, we invite you to submit a revised version of the manuscript that addresses the points raised during the review process.

**ACADEMIC EDITOR: ****Dear Authors,****one expert in the field revised your manuscript, detecting several minor points you should consider during the revision process.** Please submit your revised manuscript by Nov 18 2024 11:59PM. If you will need more time than this to complete your revisions, please reply to this message or contact the journal office at plosone@plos.org. Please include the following items when submitting your revised manuscript:

We look forward to receiving your revised manuscript.

Kind regards,

Emiliano Cè

Academic Editor

PLOS ONE

**Journal Requirements:**

This work was supported by the National Institute of Advanced Industrial Science and Technology (AIST) and the Moonshot R&D Program under Grant JPMJMS2239.

6. We note that Figure 4a includes an image of a participant in the study. 

Reviewers' comments:

Reviewer's Responses to Questions

**Comments to the Author**

1. Is the manuscript technically sound, and do the data support the conclusions?

Reviewer #1: Yes

2. Has the statistical analysis been performed appropriately and rigorously? 

Reviewer #1: Yes

3. Have the authors made all data underlying the findings in their manuscript fully available?

Reviewer #1: Yes

4. Is the manuscript presented in an intelligible fashion and written in standard English?

Reviewer #1: Yes

5. Review Comments to the Author

**Reviewer #1:** 1) Kindly include recent articles that can be related to this study and elaborate the impact of this study to current practices in related industry/research field.

2) Incorporating a discussion on how the simulation accounts for biological variability (e.g., age, gender, muscle size, and fatigue levels) would be valuable. Muscle dynamics can vary significantly across individuals, so addressing how the framework adapts or can be tuned to different populations would enhance its applicability in broader contexts such as clinical diagnosis or rehabilitation

3) for the experimental session, is there any quality control that have been adapted and by which studies/research?Additionally, including explanation on limitations—such as handling noise in sEMG signals, the complexity of muscles system, or challenges would provide a more balanced view and set expectations for future research.

4) Can the framework handle larger-scale simulations involving multiple muscle groups or more intricate movements? Including a roadmap or future directions for expanding the model’s capabilities would be helpful for readers considering its long-term potential.

6. PLOS authors have the option to publish the peer review history of their article (what does this mean?). If published, this will include your full peer review and any attached files.

Reviewer #1: No

---

## [Author Response · Author response to Decision Letter 1]

21 Oct 2024

Reviewer #1: 1) Kindly include recent articles that can be related to this study and elaborate the impact of this study to current practices in related industry/research field.

Authors' Response: We have updated the introduction section to include a discussion on the current scientific debate surrounding the effects of various neuromuscular and biological parameters on the frequency domain of sEMG signals. This provides context for how our model fits into and contributes to the ongoing conversation. Additionally, we have expanded on this in the discussion section (Future Plans subsection), where we outline how the proposed model will be used in future research to help clarify these unresolved issues. To support this, we have added relevant recent references ([11, 12, 13, 61, 62, 63]) that highlight both the debate and the current efforts within the scientific community to address these questions.

Text in the Introduction:

The complexity of the motor control system and current technical limitations make

it impossible to simultaneously measure both sEMG signals and all the neuromuscular

parameters that produce them. This has led to various discussions within the scientific

community regarding the feasibility of inferring neuromuscular parameters from the

evaluation of sEMG signals [11–13]. The model presented in this paper enables the

creation of a programmatic environment in which both neuromuscular parameters and

the generated sEMG signal are known, providing a reasonable framework for testing the

relationships between them.

Text in the Discussion:

In recent years, several studies have emerged demonstrating the potential for decomposing the spectrum of sEMG signals into two distinct components, whose behavior seems to represent groups of motor unit action potentials with varying frequencies [61, 62]. In 2022, the authors of this paper evaluated the evolution of these components during the process of muscle fatigue [63]. The results led to the hypothesis of a potential correlation between these components and the distribution of active muscle fiber types. This remains a topic of ongoing debate [11–13], as it is known that sEMG frequencies are influenced by numerous factors that cannot be simultaneously measured during sEMG recordings.

One of the primary motivations for developing this model was to establish a reference framework for linking the characteristics of sEMG signals to both internal neuromuscular parameters, and electromagnetic attenuation and dispersion factors, enabling the study of the influence of each one of them on sEMG frequencies. The results of this study demonstrate significant dependencies on the distribution of muscle fiber types in the median frequency of the simulated signals (see Fig. 7). In future works, author will aim to replicate the spectral decomposition reported in real signals using simulated sEMG signals, and then explore correlations between these components and the proportions of active muscle fibers. The ability to estimate the distribution of muscle fiber types using non-invasive methods could lead to significant advancements in digital health, facilitating the development of personalized training strategies for athletes, the early detection of muscular disorders such as sarcopenia and the overall monitoring of muscle condition.

2) Incorporating a discussion on how the simulation accounts for biological variability (e.g., age, gender, muscle size, and fatigue levels) would be valuable. Muscle dynamics can vary significantly across individuals, so addressing how the framework adapts or can be tuned to different populations would enhance its applicability in broader contexts such as clinical diagnosis or rehabilitation.

Authors' Response: Thank you for your valuable suggestion. Indeed, the highly parametric nature of our model allows it to be adapted to account for biological variability, provided that empirical data is available for tuning the model parameters. We have added a paragraph at the end of the "Model Novelty" subsection to highlight how the framework can be adjusted to accommodate differences in age, gender, muscle size, fatigue levels, and other individual-specific factors. This addition enhances the applicability of the model in broader contexts such as clinical diagnosis and rehabilitation.

Text in the Discussion:

The fully parametric nature of the proposed model allows it to account for biological variability across populations, such as differences in age, gender, and muscle structure. By adjusting the parameters of each neuromuscular component (e.g., muscle fibers, motor neurons, recruitment strategies), the model can be configured to reflect the specific characteristics of any individual, as long as these values are known. This adaptability makes the framework applicable to personalized scenarios, such as tailored rehabilitation or clinical diagnosis, where individual-specific data is available.

3) for the experimental session, is there any quality control that have been adapted and by which studies/research?Additionally, including explanation on limitations—such as handling noise in sEMG signals, the complexity of muscles system, or challenges would provide a more balanced view and set expectations for future research.

Authors' Response: This is an important point that we carefully considered during the experimental design. While noise removal filtering is commonly used in sEMG data processing, we deliberately avoided it in this study as it would alter the overall spectral distribution of the signals. Since our objective was to compare real data with simulated sEMG, which did not include noise sources, we focused on minimizing noise at the source rather than relying on post-processing.

To achieve this, we employed the Biodex system, which allowed for body and arm fixation to minimize motion artifacts in the biceps area. Additionally, the experiment was conducted in an isolated room to reduce the influence of white Gaussian noise on the higher frequency ranges of the signals, and wireless electrodes were used to prevent power line interference. This information has been added to the Experimental Protocol subsection.

As an additional quality control step, we evaluated the spectral distribution of all recorded data post-experiment to ensure that only the expected sEMG bandwidth was present, safeguarding the integrity of the data for comparison with simulated signals. We have also referenced [51], which characterizes common noise sources in sEMG, and included this in the Data Curation subsection.

Text in Experimental Protocol

The participant's body was secured to the chair with a strap, and an elbow support was used to minimize any movement in the bicep’s region, except for natural muscle contraction during tasks. These precautions were taken to prevent low-frequency motion artifacts from affecting the sEMG signals. Additionally, the experimental room was isolated to minimize the coupling of white Gaussian noise, and wireless sEMG electrodes were used to avoid power line interference.

Text in Data Curation Subsection

The spectral distribution of the measured sEMG signals was carefully evaluated to ensure the absence of common artifacts such as white Gaussian noise in the higher frequencies, power line interference at 50/60 Hz, and motion artifacts in the lower frequencies [51].

4) Can the framework handle larger-scale simulations involving multiple muscle groups or more intricate movements? Including a roadmap or future directions for expanding the model’s capabilities would be helpful for readers considering its long-term potential.

Authors' Response: The current model is capable of simulating any individual muscle, provided that MRI scans or other 3D representations of muscle and biological tissue are available. However, the model does not yet incorporate skeletal biomechanics, and the simulated muscles do not physically contract, meaning it does not currently handle full kinematics.

The framework can be expanded to include these features, though the high level of parameterization would lead to significant computational demands. Our immediate focus is to use the model as it stands to clarify several ongoing debates in the scientific community, as mentioned in the response to comment 1. Once this phase is complete, we plan to undertake code refinement and optimization to explore how computational requirements can be reduced.

If successful in reducing these demands, the next steps will involve simulating physical muscle contractions. Following this, we will explore the activation of multiple muscles in coordinated movements, which will require enhancements to the motor control system. Specifically, it will need to manage the temporal recruitment of both synergistic and antagonistic muscles, in addition to motor unit recruitment within individual muscles. Achieving this will allow us to integrate our model with higher-level motor coordination hypotheses, such as muscle synergies, and enable simulations of more intricate movements involving multiple muscle groups.

This is an important future projection for our model, and we have included in the discussion section. We would like to thank the reviewer for this suggestion.

Discussion Section (Future Plans Subsection)

Furthermore, the model holds potential for expansion towards simulating more complex muscle activations. In the future, one of the key objectives will be to implement physical muscle contractions within the model, which would allow it to simulate kinematics. This would involve incorporating skeletal biomechanics and enhancing the motor control system to manage the recruitment of multiple muscle groups in coordinated movements. Such an expansion will also require improvements to the motor control module to manage the temporal recruitment of both synergistic and antagonistic muscles. If these enhancements are achieved, it will be possible to integrate our model with high-level motor coordination theories, such as muscle synergies, enabling simulations of more intricate movements involving multiple muscle groups.

Final Authors’ Comment: We would like to sincerely thank the reviewer for their thoughtful comments and valuable suggestions. Your feedback has helped us improve the clarity and quality of our manuscript, and we greatly appreciate the time and effort you have dedicated to reviewing our work. We hope that the revisions we have made address your concerns and contribute to the overall strength of the paper.

---

## [Decision Letter · Decision Letter 1]

29 Jan 2025

Tailoring Neuromuscular Dynamics: A Modeling Framework for Realistic sEMG Simulation

PONE-D-24-29843R1

Dear Dr. Costa Garcia,

We’re pleased to inform you that your manuscript has been judged scientifically suitable for publication and will be formally accepted for publication once it meets all outstanding technical requirements.

Kind regards,

Emiliano Cè, Ph.D.

Academic Editor

PLOS ONE

Additional Editor Comments (optional):

Reviewers' comments:

Reviewer's Responses to Questions

**Comments to the Author**

1. If the authors have adequately addressed your comments raised in a previous round of review and you feel that this manuscript is now acceptable for publication, you may indicate that here to bypass the “Comments to the Author” section, enter your conflict of interest statement in the “Confidential to Editor” section, and submit your "Accept" recommendation.

Reviewer #1: All comments have been addressed

2. Is the manuscript technically sound, and do the data support the conclusions?

Reviewer #1: Yes

3. Has the statistical analysis been performed appropriately and rigorously? 

Reviewer #1: Yes

4. Have the authors made all data underlying the findings in their manuscript fully available?

Reviewer #1: Yes

5. Is the manuscript presented in an intelligible fashion and written in standard English?

Reviewer #1: Yes

6. Review Comments to the Author

Reviewer #1: (No Response)

7. PLOS authors have the option to publish the peer review history of their article (what does this mean?). If published, this will include your full peer review and any attached files.

Reviewer #1: No

---

## [Editor Report · Acceptance letter]

PONE-D-24-29843R1

PLOS ONE

Dear Dr. Costa-Garcia,

I'm pleased to inform you that your manuscript has been deemed suitable for publication in PLOS ONE. Congratulations! Your manuscript is now being handed over to our production team.

Kind regards,

on behalf of

Prof. Emiliano Cè

Academic Editor

PLOS ONE